# Health Risks and Consequences of a COVID-19 Infection for People with Disabilities: Scoping Review and Descriptive Thematic Analysis

**DOI:** 10.3390/ijerph18084348

**Published:** 2021-04-20

**Authors:** Sureshkumar Kamalakannan, Sutanuka Bhattacharjya, Yelena Bogdanova, Christina Papadimitriou, Juan Carlos Arango-Lasprilla, Jacob Bentley, Tiago S. Jesus

**Affiliations:** 1Public Health Foundation of India (PHFI), South Asia Centre for Disability Inclusive Development and Research (SACDIR), Indian Institute of Public Health-Hyderabad (IIPH-H), Hyderabad 500 033, India; Suresh.Kumar@lshtm.ac.uk; 2Department of Occupational Therapy, Byrdine F. Lewis College of Nursing and Health Professions, Georgia State University, Atlanta, GA 30303, USA; sbhattacharjya@gsu.edu; 3Physical Medicine & Rehabilitation Service, VA Boston Healthcare System, Boston, MA 02130, USA; bogdanov@bu.edu; 4Department of Psychiatry, Boston University School of Medicine, Boston, MA 02118, USA; 5Departments of Interdisciplinary Health Sciences, School of Health Sciences, Oakland University, Rochester, MI 48309-4452, USA; cpapadimitriou@oakland.edu; 6IKERBASQUE, Basque Foundation for Science, 48903 Bilbao, Spain; jcalasprilla@gmail.com; 7Biocruces Bizkaia Health Research Institute, 48903 Barakaldo, Spain; 8Department of Cell Biology and Histology, University of the Basque Country UPV/EHU, 48903 Leioa, Spain; 9Department of Clinical Psychology, Seattle Pacific University, Seattle, WA 98119, USA; bentley@spu.edu; 10Department of Physical Medicine & Rehabilitation, Johns Hopkins School of Medicine, Baltimore, MD 21205-2196, USA; 11Global Health and Tropical Medicine (GHTM) & WHO Collaborating Centre for Health Workforce Policy and Planning, Institute of Hygiene and Tropical Medicine, NOVA University of Lisbon, 1349-008 Lisbon, Portugal; 12Department of Occupational Therapy, College of Health & Rehabilitation Sciences: Sargent College, Boston University, MA 02215, USA

**Keywords:** COVID-19, health equity, disabled persons, vulnerable populations, public health

## Abstract

This study aims to synthesize the literature on any disproportionate health risks or consequences of a COVID-19 infection for people with disabilities. Scoping review with a descriptive thematic analysis was carried out. Up to mid-September 2020, seven scientific databases and three preprint servers were searched to identify empirical or perspective papers. Snowballing searches and expert’ consultations also took place. Two independent reviewers were used for the screenings and data extractions. Of 1027 references, 58 were included, 15 of which were empirical articles. The thematic analysis showed that: (1) People with disabilities living in residential or long-term care facilities were more likely to have greater infection rates; (2) Intersecting mediators of greater infection risks were multiple (e.g., lack of accessible information); (3) People with disabilities often face greater health problems when infected; and (4) Unethical disadvantages in the rationing of lifesaving and critical care can be experienced by people with disabilities. Conclusions: Beyond any health-related vulnerabilities (e.g., comorbidity rates), multiple yet modifiable environmental factors can provide disproportionate health risks and consequences of a COVID-19 infection for people with disabilities. Public health and policy measures must prevent or reduce modifiable environmental risks.

## 1. Introduction 

The Corona Virus Disease 2019 (COVID-19) refers to a recent infectious disease that has been causing major public health and economic crises worldwide. However, during the pandemic, concerns have been raised that the health and socio-economic impact of the pandemic may hit harder on some strata of the population, especially the most socially vulnerable (e.g., low-income, institutionalized, incarcerated, refugees, those without access to soap or clean water), and thereby can exacerbate existing health and social disparities [1,2,3,4,5].

People with disabilities refer to a group of people vulnerable to discrimination, marginalization, and multiple social disadvantages [6,7]. These include vastly documented disparities experienced in health, healthcare access, and healthcare quality and outcomes, observed before the pandemic [8,9,10]. Historically, ‘ableism’ has dominated the way people with disabilities have been perceived, including legal, political, educational practices, and healthcare discourses that exclude and/or discriminate against people with disabilities, and which result in disvaluing their lives [11]. These systematic values can put health equity for people with disabilities in jeopardy during a pandemic crisis and unethically interfere with any medical rationing decisions [12,13]. All accounted, COVID-19 has amplified existing structural inequalities in health and the social determinants of health that many people with disabilities have been facing for a long time [7,12].

People with disabilities are approached here as a ‘vulnerable group’ not because people with disabilities are inherently vulnerable (which could enforce, and misleadingly so, an ideal of normalcy), but mostly due to underlying ableist structural factors that result in vulnerability, often arising from disability stigma, marginalization, and lack of societal accommodations. These include the lack of disability-inclusive public health and health systems’ responses. Indeed, the lack of disability inclusion in society and in public health to design equitable healthcare policies is well documented by now [1,4,13,14]. Overall, aligned with others [6,7,15,16,17], this literature review is developed from the standpoint that public health and policy responses to the COVID-19 pandemic should be disability-inclusive [18].

It is expected that those special vulnerabilities experienced by people with disabilities during the COVID-19 pandemic could result in greater exposure to or risks of being infected, not having timely access to a COVID-19 diagnosis or treatment once infected or experiencing more frequent or severe health and socio-economic consequences, either being infected or not. Within the context of these possibilities, we have planned to develop a scoping review of the literature addressing any, and often multiple, vulnerabilities faced by people with disabilities during the COVID-19 pandemic [18].

This paper refers to the first part of those scoping review results. Here, we aim to synthesize the literature on any disproportionate health risks or consequences of a COVID-19 infection for people with disabilities. By disproportionate health risks, we mean any greater risks for people with disabilities to be infected with the COVID-19/SARS-COV-2. By disproportionate health consequences, we mean any greater likelihood of having more severe health consequences once being infected. The latter can arise from any individual vulnerability (e.g., comorbid conditions) or the lack of equitable and timely access to healthcare. 

## 2. Methods

A scoping review methodology was used, following the methods described in the published, open-access scoping review protocol [18]. In this paper, descriptive thematic analysis, later detailed, is used to understand any disproportional health risks or consequences of a COVID-19 infection for people with disabilities. In turn, a second results paper [19] addresses the lockdown-related health and socio-economic disparities that people with disabilities may experience during the COVID-19 pandemic, without necessarily being infected. The Preferred Reporting Items for Systematic reviews and Meta-Analyses extension for Scoping Reviews (PRISMA-ScR) were used to guide this report [20]. 

### 2.1. Eligibility Criteria

We included peer-reviewed empirical or perspective papers (including editorials or commentaries) or pre-print empirical studies explicitly addressing: (1) the COVID-19 disease or pandemic; (2) People with disabilities as a group, subgroup (e.g., based on impairment type or underlying diagnostic condition), or related individual circumstances as a pre-condition—i.e., excluding impairments arising only as a consequence of the COVID-19 infection; and (3) an individual-level (e.g., health- or age-related) or social-level (e.g., living condition, and income-related) vulnerability to a COVID-19 infection or its health consequences. 

This open-access study protocol provides working definitions of people with disabilities and vulnerability, including a text-box with the possible individual (e.g., comorbidities) and social-level vulnerabilities (e.g., suboptimal access to healthcare, and living in institutional settings) to the effects of the COVID-19 pandemic, as a means to support eligibility decisions [18]. There were not any geographic restrictions and papers in 6 languages (i.e., English, French, Spanish, Greek, Russian, and Portuguese) were considered; however, although articles in languages other than English were detected by the searches, none fully met the other eligibility criteria. 

### 2.2. Information Sources and Search 

Seven databases for the scientific, peer-reviewed literature (i.e., Medline/PubMed, Web of Science–Core Collection, Scopus, AgeLine, PsycINFO, CINAHL, and ERIC) were used to search. Searches were run in mid-July 2020 and repeated two months later, as described in the study protocol [18]. Hence, we cover the literature up to mid-September 2020, which roughly equates to data and perspectives from the first wave of the COVID-19 pandemic. Three databases for preprint literature (i.e., MedRxiv, SocArXiv, and PsyArXiv) were also searched, following the same process and dates. During the initial stages of the COVID-19 pandemic, preprint databases have been hosting many studies that have not yet reached peer-reviewed publications [21]. Furthermore, this option can help to avoid the exclusion of studies with negative results which may be published less often or less rapidly. Before the data charting, we searched for the peer-reviewed version of the included preprints and have replaced the record whenever found. The full search strategies for each of the scientific and preprint databases were provided as a supplementary appendix in the open-access protocol [18], as here are reported in the Appendix A.

A snowballing search process (e.g., author tracking, and referenced sources) was also conducted, using the included references to identify any additional records. Supplied with a preliminary list of inclusions, members of the American Congress of Rehabilitation Medicine’s International Networking Group and Refugee Empowerment Task Force were also consulted as key informants to provide any additional references.

Although planned [18], we did not include elements of the grey literature (e.g., official reports from international organizations). During the initial searches, we have found a living repository of that literature, hosted by the United Nations. That freely accessible repository (https://www.un.org/development/desa/disabilities/covid-19.html (Accessed date: 15 December 2020)) provides key grey literature resources from the United Nations, their specialty agencies, and from partner institutions (e.g., Disabled Person’s Organizations) alike. With this new information, and to produce timely results as intended [18], we opted to exclude the grey literature and narrow the review coverage to the peer-reviewed literature and preprint studies. An iterative development process is common in scoping reviews, with some decisions—as long as justified and reported—taken as new information comes by, since scoping reviews usually explore and map out initially unchartered territories [22,23]. 

### 2.3. Selection Process

The abstract-and-titles screenings and the full-text assessments were made against the eligibility criteria and were conducted by two independent reviewers (S.K. and S.B.), after pilot screenings with over 80% agreements, overseen by the leading review author (T.J.). Any discrepancies were resolved through consensus or the leading author’s input. 

### 2.4. Data Charting and Items

Following a coding structure elaborated by members of the research team, one author (S.K.) extracted formal data elements (publication type, sources, geographies addressed), with a random 5% verified by another (T.J.). Regarding the content of the literature included, two independent reviewers (SK and SB) extracted text quotations on 1) any added risk for, or 2) disproportionate consequences of a COVID-19 infection on people with disabilities. These independent extractions were later paired for the qualitative data synthesis, which was also informed by a brief synthesis of each paper developed by two reviewers independently. Then, the Appendix A provides the content of these extractions after being merged (i.e., presented as the combined extractions of both reviewers), as well as reviewers’ combined synthesis of each paper.

### 2.5. Critical Appraisal

No methodological quality assessments were performed as described in the study protocol [18] and common in scoping reviews [22,23,24].

### 2.6. Synthesis of the Results

Simple descriptive statistics (e.g. counts, and percentages) are computed to provide a summative, tabular description of the amount and range of the related literature, including per publication type and source, country (or countries), or health conditions or impairments addressed. 

A descriptive thematic analysis was developed to synthesize the text quotations [25]. The protocol for the entire scoping review project [18] defines the use of thematic analysis in full, sometimes labeled as a reflexive thematic analysis [25,26]. However, that was essentially applied over the content presented in a subsequent paper [19], addressing the more complex intersections among health, social, economic, or occupational injustices or inequalities experienced by people with disabilities due to lockdown-related measures. For the issues being reviewed here, we apply a descriptive thematic analysis, as we did not formulate a new interpretive schema but essentially provided a thematic clustering of the findings—for any disproportionate health risk or health consequence of a COVID-19 infection among people with disabilities. 

Finally, as described in the study protocol [18], we took a final consultation stage. Supplied with a preliminary version of the results and its discussion, members of the American Congress of Rehabilitation Medicine’s International Networking Group and Refugee Empowerment Task Force had the opportunity to comment and provide improvement suggestions over the preliminary results and their interpretation. 

## 3. Results

Figure 1 provides the flowchart of this review. Out of 1027 unique references, 58 are finally included in this analysis, i.e., report findings or rationales for any disproportionate health risks or health consequences of a COVID-19 infection for people with disabilities. The Appendix A lists the 58 papers included. 

Table 1 reflects the distribution of the analyzed articles by publication type and source, by geographical focus, and finally by the health conditions or impairments addressed. 

Among the 58 papers included, 15 were empirical studies (two of which preprints), and the vast majority (73%) were non-empirical (e.g., perspective papers). Close to two-thirds (63%) of the papers had no geographical focus (e.g., were applicable across locations). When they had a geographical focus, most (12 out of 19) addressed the United States of America (USA) or the United Kingdom (UK) context. More than half of the papers (56%) addressed people with disabilities overall (i.e., had no focus on specific health conditions or impairments). 

In the sections below, we provide the descriptive thematic analysis of any disproportionate health risks or consequences experienced by people with disabilities. 

### 3.1. People with Disabilities Living in Residential or Long-Term Care Facilities Were More Likely to Have Greater Infection Rates

Greater COVID-19/SARS-CoV-2 infection rates among people with disabilities living in residential or long-term care facilities were reported by included studies or review of studies.

Using data for people with intellectual and developmental disabilities from a coalition of organizations providing over half of the residential services for the state of New York, and from the New York State Department of Health, the COVID-19/SARS-CoV-2 case rate for people with intellectual and developmental disabilities was found to be over four times greater than for the New York State general population, likely due to the high percentage of those who live in congregated care settings [70]. However, the interpretation of these numbers needs to account for the incipient health surveillance data focused on people with intellectual and developmental disabilities, which can result in limited reporting on COVID-19 outcomes for this population [70].

Similarly, a narrative review of overall health inequities has found evidence showing that people with disabilities living in residential facilities can have a five times greater likelihood of being infected by the SARS-CoV-2 than the general population [59]. In residential facilities for people with severe epilepsy, enhanced surveillance revealed high rates of asymptomatic SARS-CoV-2 infected residents (from 56 to 78% in varying residencies), for a relatively low (9%) symptomatic infection rate; this finding highlights how easily the infection can spread in these contexts, including through asymptomatic cases [78]. Finally, a literature review found a high rate of dementia in cases of people hospitalized with COVID-19 (values varied from 6.8 to 13.1% across studies) [71], while the WHO estimates a 5–8% prevalence of dementia for the population aged over 60. 

For people with disabilities living in the community, the few and only partly related existing data pointed in an opposite direction. In a pre-print ecological study of the non-institutionalized disabled population, within the most affected counties of the United States and up to 9 April 2020, the bivariate regression analyses indicate that counties with a higher White disabled population (95% CI: −0.43-(−0.02); *p*-value: 0.03), higher population with hearing disability (95% CI: −0.42- (−0.11); *p*-value: 0.001.2), and higher population with disability in the 18−34 years age group (95% CI: −0.41-(−0.09); *p*-value: 0.002) had a lower rate of SARS-CoV-2 infection—yet higher mortality rates as later depicted [77]. However, these data are from an ecological study with the main unit of analysis on counties and their population strata rather than on people with disabilities themselves.

### 3.2. Intersecting Mediators of Greater Infection Risks 

The literature reviewed revealed a myriad of causes explaining why people with disabilities may experience greater infection risks, and notably far greatest infection rates observed among institutionalized people with disabilities. 

In residential care settings, shielding and self-isolation were difficult during the first wave of the pandemic. This was due to the: initial unpreparedness of the facilities at the pandemic outset, shared use of essential living spaces, crowding and shared rooms, proximity to other residents, residents’ difficulty understanding new rules imposed suddenly, difficulty in maintaining standards of hygiene during home visits, multiple shift staffing patterns, staff working in multiple settings, and high levels of personal care assistance required from staff (e.g., with eating, toileting, or transferring from bed to wheelchair) [16,30,31,36,38,39,43,51,53,54,59,70,78]. Indeed, stemming from the latter reason, many articles pointed out that a wide range of people with disabilities who rely on assistants for basic activities of daily living have greater risks of infection [6,32,36,41,49]. 

Regarding impairment subgroups, people with intellectual, developmental, or cognitive impairments (e.g., from dementia), either institutionalized or not, might not understand, remember, or be able to systematically comply with the quarantine or other preventive measures (e.g., use of masks, washing hands, and physical distancing), and may even respond with challenging behavior including motor agitation, intrusiveness, or wandering, which may undermine efforts to maintain isolation and other preventive measures [36,40,42,52,55,62,71,78]. 

Among persons with severe mental illness, 8 to 23% were not aware of the ongoing COVID-19 pandemic, the need for quarantine, or the mode of spread of COVID-19, either due to cognitive impairments or lack of access to information from the media; therefore, unlike the general population, about 75% of the persons with severe mental illness did not report fear of contracting COVID-19 and may not have taken protective measures [66]. People with intellectual and cognitive disabilities may have difficulties identifying or communicating that they are experiencing mild symptoms, and this can help justify the high rates of seemingly asymptomatic cases observed in a residential setting for people with disabilities [78].

People with visual or hearing impairments also may face infection risk because of a lack of accessible information on preventive measures during the initial stages of the pandemic. [16,48,49,50] In Low- and Middle-Income Countries (LMICs), the situation can be worse [49]. For instance, an audit of press conferences in LMICs showed that only 65% of countries have used a sign language interpreter, with figures varying from 33% to 88% [74]. Globally, another analysis showed that pages of the World Health Organization’s website, March–May 2020, were only 60% compliant regarding web accessibility guidelines [75]. Additionally, people with sensory processing and visual impairments may need to rely or depend on touch and tactile senses for stimulus or to perform their routine activities or outdoor movement, and thereby face greater infection risks [33,38,40,43,49].

For people with Down Syndrome, increased cytokine production can make them vulnerable to contracting infections like COVID-19, with insufficient immunity after disease recovery and a possible higher risk of reinfection, or even lower immune response to vaccination [76].

Focusing on environmental circumstances, in Latin America, millions populate densely packed ‘favelas’ in which large families often share a single room, and many consider moving a grandparent to a nursing home inconceivable; hence, older people and those with disabilities are quarantined within crowded living quarters, where they can be exposed to asymptomatic carriers [64]. Moreover, in LMICs, many people with disabilities lack access to clean water, live in crowded institutional settings, and have added difficulties complying with hand hygiene, physical distancing, and other protective measures [38]. 

Children with disabilities often rely heavily on ride-sharing programs for transportation to school and other activities; these modes of transport compel more exposure than private transportation [27]. Prevalence of disabilities is also high for incarcerated people, where rates of COVID-19 infection and transmissions have been great [3].

Finally, the results of survey research in Italy may help explain why some groups of people with disabilities living in the community may have been less exposed to a COVID-19 infection. The study showed generally good adherence of persons with Multiple Sclerosis (*n* = 551) to lockdown and extensive use of protection devices, as measured by Lockdown Scores [65]. Overall, the authors found that women, older persons, persons with disabilities, and those who were unemployed had higher Lockdown Scores (*p* < 0.05) [65]. It is possible though, that this can arise from the combination of greater cautions due to the fear of the greater health consequences of an infection as well as lower social participation levels. 

### 3.3. People with Disabilities often Face Greater Health Consequences When Infected

In a study using a large dataset from electronic medical records of a COVID-19 Research Network, the case-fatality rate was much higher for people with intellectual and developmental disabilities compared to non-disabled counterparts at younger ages such as <17 (1.6% versus 0.01%) and 18 to 74 years (4.5% versus 2.7%), even though similar (5.4% versus 5.1%) for all ages [69], possibly reflecting greater prevalence of comorbidities among people with disabilities relative to non-disabled counterparts at younger ages [69]. 

In one ecological study of the associations between infection and mortality rate of COVID-19 and demographic, socioeconomic, and mobility variables from 369 counties in the USA, a higher rate of people with disability in the county was significantly associated with higher mortality (95% CI: 0.09–0.45) [72]. In turn, the same study found that while poverty and disability are frequently associated, their interaction was not significant (*p* = 0.469) and could have had independent contributions as risks to mortality [72]. 

In New York state, a case-fatality rate of 15% among people with intellectual or developmental disabilities was about double of the general population [70]. In residential facilities for people with epilepsy in the UK, the case fatality rate was high (50% or 11% corrected for asymptomatic) [78]. An ecological study found significant correlations between Disability-Adjusted Life Years from dementia with COVID-19 mortality [71], which is aligned with the results of underlying literature reviewed indicating that pre-existing dementia was significantly associated with COVID-19 severity and increased mortality rate [71].

People with rheumatoid arthritis are intrinsically characterized by an increased infectious risk due to the disease itself and to the iatrogenic effect of immunosuppressive agents such as corticosteroids [60]. In persons with Down Syndrome, increased cytokine production (e.g., COVID-19 mortality also accounts for cytokine release syndrome), more vulnerable immune systems, and frequent comorbidities like diabetes, heart, and respiratory conditions, can all make these persons vulnerable to mortality and other deleterious consequences of infections like COVID-19 [34,35,76]. Individuals with Cerebral Palsy also frequently experience comorbidities such as pulmonary disease or elevated blood pressure [29]. In people with cerebellar ataxia, dysphagia, or ataxic respiration, the maintenance of airway protection, and aspiration pneumonia are ever-present concerns, with COVID-19 infections adding to pulmonary complications [67].

For individuals with spinal cord injury, several special considerations apply. Physiological changes not only increase their risk of morbidity from COVID-19 but may also mask the presentation of acute respiratory illnesses and delay the diagnosis of COVID-19, while individuals with cervical or high thoracic spinal cord injuries may not have chest sensations or the ability to cough [37,58,63]. In an international survey of healthcare professionals who care for those with spinal cord injury (*n* = 783), 10.3% reported their patients with COVID-19 had increased spasticity and 6.9% reported that their patients had rigors [68]. 

Patients with dementia who develop COVID-19 also may present atypical symptoms, such as delirium, rather than respiratory symptoms, which also affect the early recognition [71]. In turn, people with intellectual disabilities overall might be unable to identify or report infection symptoms [31,61]. 

Individuals with impaired mobility, including people with visual impairments, may not be able to access drive-through COVID-19 test facilities for early detection of the disease [49]. In turn, people with intellectual disabilities, as well as deaf and hard-of-hearing individuals, are unable to use many of the complex, non-inclusive automatic answering systems to access COVID-19 screenings [49]. Furthermore, the non-visitor policies at treating hospitals might impede a caregiver-mediated information exchange and shared decision-making on critical care decisions for people with intellectual disabilities, while structural issues such as inaccessible facilities and equipment, existing for decades, impede optimal COVID-19 diagnostic and treatment for people with disabilities [49]. In China, during the COVID-19 quarantine, one disabled teenager died after being left at home for six days without care, while his relatives were quarantined, which can illustrate how needed acute healthcare may be inaccessible for unsupported people with disabilities [32].

Overall, in addition to any associated comorbidities, poorer outcomes for people with disabilities can come from difficulty to detect or communicate the infection symptoms as well as from the restricted or delayed access to essential public health information and life-saving healthcare [15,16,30,31,36,42,54,61,72,77].

### 3.4. The Unethical Disadvantage in the Rationing of Lifesaving and Critical Care

In a systematic review of Crisis Standards of Care across USA states, it was found that only two-thirds explicitly articulated that resource allocation decisions should be made without regard to race, ethnicity, disability, and other identity-based factors [73]. The same review found that low likelihood of immediate survival was frequently stated for exclusion, including examples such as “advanced” and/or “irreversible” neurologic events or “severe dementia” (70% of states had these exclusion criteria) [73], while a neurologic disorder (e.g., Alzheimer’s dementia) was one the most frequently cited health conditions to be considered for low prioritization decisions (33.3%) [73]. 

Other published accounts point out that, in the USA, some states were endorsing guidelines to withhold ventilators and treatment from those with certain neurological impairments and intellectual disabilities [53,56,59,62]. In some jurisdictions, policies have been issued for people with disabilities to have a lower priority to receive critical and lifesaving medical care; at a very minimum, confusion around this (or lack of clear statements otherwise) could further limit access to lifesaving care for people with disabilities [6,28,44,46,48,53,56]. For example, guidelines stating that survival of younger healthy persons is to be prioritized relative to “chronically debilitated patients” can be interpreted to disadvantage people with disabilities, even if young or otherwise healthy [53]. Finally, beyond ethically questionable, explicitly excluding people based on disability and is probably illegal under anti-discrimination laws [56].

Medical rationing decisions based on markers such as ‘quality of life’ may lead healthcare providers to discriminate against disability due to preconceived (and often without awareness) ableist notions about people with disabilities’ quality of life [13,44,47,53,56]. Physicians and health care providers may well be influenced by implicit and explicit ableism [39]. Stereotypes about what life is like living with a disability can be improperly used to exclude people from needed care [35,49]. Healthcare providers in the field, forced to make rapid decisions, sometimes in the absence of guidelines, will inevitably bring societal and personal ableist biases—with or without awareness—into the triage process which, if unchecked, may result in stark inequities in both care and survival for people with disabilities at the population level [73]. Ethical issues surrounding priority vaccination and treatments are likely to affect the health impact of the COVID-19 in the disability community [34]. 

Some guidelines for access to critical care would imply, or be interpreted, as a form of disablism or social utilitarian perspective, by directly or indirectly excluding (younger) people with disabilities but otherwise healthy and no less likely than other people of the same age to recover with treatment, when impairments per se often have no impact on the likelihood of survival [12,56]. Furthermore, people with disabilities that require permanent ventilation to live may be threatened by the lack of ventilators to respond to the pandemic [27,57].

The pandemic is prompting healthcare providers to think more about rationing, which may result in the devaluing of the lives of patients with compromised decision-making capacity (e.g., cognitively impaired) [45]. Furthermore, in the context of a pandemic, time and other pressures on healthcare providers may impede their ability to carry out capacity assessments in full, notably to judge whether one person has the cognitive capacity to take or participate in critical medical decisions [51]. People with intellectual disabilities may become subject to rationing and the non-use of potential lifesaving treatments, as merely assumed by providers [61].

In South Africa, accounts of utilitarian COVID-19 triage policies and practices have been found, disadvantaging many people with disabilities, especially people with physical and intellectual impairments, from gaining intensive care unit access, while not providing care on a utilitarian basis, e.g., factoring in the patient’s cognitive impairment is argued to be against the United Nations Convention on the Rights of Persons with Disabilities (UN-CRPD) [38,51]. In the UK, there was a higher threshold for frail elders and those with so-called challenging behavior in long-term care to be admitted to hospitals, as well as their access to intensive care was restricted [78]. 

Overall, during the first wave of the pandemic, people with disabilities were worse-off compared to their non-disabled peers in terms of risk factors for more severe outcomes and faced an often inaccessible healthcare system with a history of ableism and discrimination against people with disabilities. In particular, there is an absence of consolidated guidelines for emergency responses that are explicitly free of disability stigma or bias [50].

## 4. Discussion

This scoping review addresses the literature published during the initial stages of the COVID-19 pandemic (up to mid-September 2020), and the results suggest that people with disabilities can experience disproportionate health risks and consequences from a COVID-19 infection. These disparities can result from several intersecting factors. On the one hand, health-related factors such as greater comorbidity rates among younger people with disabilities, compared to non-disabled counterparts, may justify the much greater fatality rates observed for that subgroup of people with disabilities. On the other hand, several environmental factors contributed to turning many people with disabilities vulnerable to a COVID-19 infection or their consequences. These include multiple exposure risks in residential or long-term facilities, the lack of accessible healthcare and information, and medical rationing decisions influenced by disability stigma. Indeed, environmental determinants of transversal nature (e.g., related to disability stigma, health information and healthcare access disparities, and lack of systemic accommodations) may help justify the fact that most papers address people with disabilities in general and not by specific conditions. While the identified environmental determinants of dispositional health risks and consequences are potentially modifiable, addressing them would promote health equity for people with disabilities. 

In residential or long-term facilities, multiple infection risk factors such as the shared use of spaces, staff working in multiple settings, high levels of personal care assistance from staff (especially for residents with disabilities), among other risk factors synthesized in this review, were coupled with settings’ initial unpreparedness to manage a pandemic, to create a ‘perfect storm’, disproportionately impacting on people with disabilities, who are overrepresented in these settings. Therefore, there is a need for disability-inclusive systematic preparedness and rapid responses to pandemic events [6,15,16,17,79], which shall include a focus on residential or long-term settings and engage people with disabilities or their representatives in the development process [7,31,32,34,38,40,80]. Research evidence and equitable perspectives on vulnerabilities faced by people with disabilities, such as the ones here synthesized, can also inform the development of disability-inclusive responses and preparedness plans—to a pandemic or overall disaster, crisis, or emergency events. Future plans for a disability-inclusive preparedness and response might also be designed to account for (e.g., include counter-measures for preventing or mitigating) any unintended consequences of the infection prevention measures. These may include social distance and loneliness, confusion, lack of (physical) activity, unmet personal needs, among other lockdowns- or quarantine-related vulnerabilities experienced by people with disabilities during the COVID-19 pandemic. These were further synthesized in a second paper resulting from this scoping review project [19]. 

The lack of accessible information (e.g., in press conferences, public health websites) was another modifiable environmental factor identified as a barrier to health equity for people with disabilities. This can be addressed with the straight application of existing guidelines, e.g., web-accessibility guidelines [75], the presence of sign language interpreters, and the use of transparent masks for the transmission of key public health information. Principles of health literacy [81] might also be useful for optimizing the access to and understanding of the needed preventive measures, for example, but not limited to, among people with developmental, intellectual, or cognitive impairments, as well as their caregivers. COVID-19 test facilities and automatic answering systems to access COVID-19 screenings might also be designed or modified to be accessible by all, including people with disabilities. 

The unethical disadvantage in the rationing of lifesaving and critical care, associated with disability stigma or stereotypes of poor quality of life among people with disabilities, was another major environmental barrier synthesized from the reviewed literature. This barrier can reflect socially entrenched misconceptions, discrimination, and prejudice toward people with disabilities [7,15,80], and was operationalized during the initial pandemic time through triage guidelines and medical decisions at least implicitly biased against people with disabilities. While the reviewed literature reports that some triage guidelines explicitly state that medical decisions should not be based on factors such as a presence or absence of a disability, many others either fail to address it or at least implicitly suggest otherwise [73]. When not explicitly stated that disability should not be equated as a factor, ableist preconceptions of medical staff may result in judging the value of a people with disabilities as one least worth living. It is important to acknowledge that impairments per se are not akin to comorbidities and that the former might not objectively interfere with chances of survival from a COVID-19 infection [12,56,57]. 

Medical decisions and triage guidelines explicitly free of disability stigma is an ethical and legal matter, i.e., both the ‘right’ and lawful thing to do [38,51,56]. More recently, ethical and legal issues can be posed around the issue of who, including people with disabilities, shall be prioritized for COVID-19 vaccination [82]. The UN-CRPD serves as a human rights ground for anti-discrimination laws and care directives. Aligned with this, for example, the Office for Civil Rights in the US has issued guidance to States and healthcare providers, early during the pandemic, stating that disability rights and anti-discrimination laws are not waived during the pandemic [83]. Furthermore, in person or remotely involving diverse people (e.g., disability activist or scholars, ethicists, healthcare providers with disabilities, rehabilitation professionals, and others more likely aware of disability rights), in reviewing or taking collective life-saving medical decisions, may reduce the likelihood of personal biases in medical rationing decisions [48,49]. For the development of equitable guidelines, people with disabilities or their advocates might be actively involved. Additionally, factors synthesized here (e.g., vulnerabilities of people with disabilities living in residential facilities, insufficient immunity in individuals with Down Syndrome) can also be taken into consideration in terms of prioritizing the most vulnerable.

Finally, we have found that few papers had a specific focus on LMICs, and even though some specific issues may apply and have been described (e.g., greater infection-spread risks through large conglomerates of people in ‘favelas’ or households), others are less well-addressed. People with disabilities living in LMICs can face multiple health and social disparities [17,84,85], and being among the least studied can also contribute to that. Further data and studies on the disproportionate impact of the COVID-19 pandemic on people with disabilities in LMICs are warranted, especially considering that health systems and structures often differ substantially from those in high-income nations and provided that the health needs of people with disabilities living in LMICs all unmet all too often [85,86,87].

### Limitations

This paper has the following limitations: First, this paper only reviewed the information available up to mid-September 2020. A continuous update of the searches was found unfeasible as the increasing trend in the number of retrieved references was outpacing our planned capacity to timely process them. We opted to review a defined timespan and not to compromise on key methods, such as the use of two independent reviewers, because eligibility and extraction decisions were not always straightforward, e.g., on the interpretation of whether any special vulnerability was involved, and the input of a third, expert reviewer was often involved. Another reason to limit the timespan was that it permitted the involvement of multiple co-authors with diverse backgrounds in the synthesis, including experts in consultation roles. Their collective input enhanced the interpretation of the findings reported in this article. Since only papers available before mid-September 2020 were included in the review, these results are limited to the initial stages of the pandemic, where the impact of unpreparedness may hit harder. Review updates, with a priori hypotheses, more focused study questions, and eligibility criteria, can eventually occur; informing more focused reviews (e.g., a systematic review) is one of the typical purposes of conducting scoping reviews in the first place [22,23]. In a rapid scan of the literature published after mid-September 2020, we detected a study detailing the deaths among people with intellectual disabilities in the UK and Ireland [88]. That study coincides with our findings and found, for instance, that younger people with disabilities can be at risk of adverse consequences, given the younger mean age of death due to COVID-19 of individuals with intellectual disability relative to the mean age of death in the general population [88].

Second, few empirical studies addressing the health risks and impact of COVID-19 infection on people with disabilities were found, which aligns with the results of a recent rapid review of the literature [89]. On the one hand, this may be due to the little time to conduct and publish studies up to mid-September 2020; in the paragraph above, we provided an example of one additional study published afterward. On the other hand, it may also reflect the fact that disability status is not systemically included in health surveillance systems or electronic medical or administrative records [34,48,54,90]. This lack of readily available data, and especially so in LMICs, may have undermined the ability to produce rapid studies on any disproportionate effects of the COVID-19 infection on people with disabilities. Furthermore, articles, in contrast to editorials, letters, commentaries, typically take more time to peer-review, which was part of the reason why preprint studies were considered in this review [21]. Designing, funding, and developing inclusive COVID-19 research is a necessity, not a luxury. Under this approach, a recent project provided a framework and checklists for addressing key issues when designing and delivering COVID-19 research, inclusive of vulnerable groups such as people with disabilities [91]. 

Third, we excluded grey literature since a key repository of that information was found (https://www.un.org/development/desa/disabilities/covid-19.html (Accessed date: 15 December 2020)) and this decision allowed for timelier results. Of note, many perspective papers included in this study had already considered and integrated the content of the grey literature. However, as the grey literature was not included, we cannot determine whether our results overall, and our themes, in particular, would have differed otherwise. Although the perspective papers included are not empirical papers, they were peer-reviewed (i.e., had scrutiny by other experts) and provided key qualitative accounts, reported experiences in some countries, discussed key ethical perspectives and threats, and other important information that contributed to these findings and provided context to help interpret the data from various studies.

Fourth, no methodological quality assessments were performed, as it is common in scoping reviews. Hence, we cannot assure whether reported study findings result from scientifically sound methodological processes. Further, up-to-date systematic reviews on specific topics here explored (e.g., increased infection or death rates among people with disabilities living in residential or long-term care settings) should provide more trustable results (i.e., involving risk-of-bias assessments) for these particular findings.

Fifth, little and somewhat inconclusive data was found on any disproportionate infection rates among people with disabilities living in the community. As an exception, one study showed that community-dwelling people with disabilities extensively used protection devices, as measured by Lockdown Scores, similar to women, older persons, and those who were unemployed [65]. It is possible that lower social participation levels of these often marginalized or vulnerable groups may temporarily protect people with disabilities living in the community from infection risks, but exacerbates other structural, social, and even health disparities (other than from a COVID-19 infection), which are addressed in our second paper [19].

## 5. Conclusions

This scoping review, addressing the initial stages of the COVID-19 pandemic, suggests that people with disabilities can experience disproportionate health risks and consequences from a COVID-19 infection. These risks and disparities are challenging and go beyond any health-related vulnerabilities (e.g., the greater presence of comorbidities among younger people with disabilities relative to non-disabled counterparts), and in fact entail multiple environmental factors. These include, for example, multiple exposure risks for people with disabilities who live in residential facilities, lack of accessible healthcare and information, and medical rationing affected by ableism and disability stigma. These environmental determinants of disparities can, and should, be modified to prevent or mitigate any disproportionate health risks and consequences of a COVID-19 infection for people with disabilities worldwide.

## Figures and Tables

**Figure 1 ijerph-18-04348-f001:**
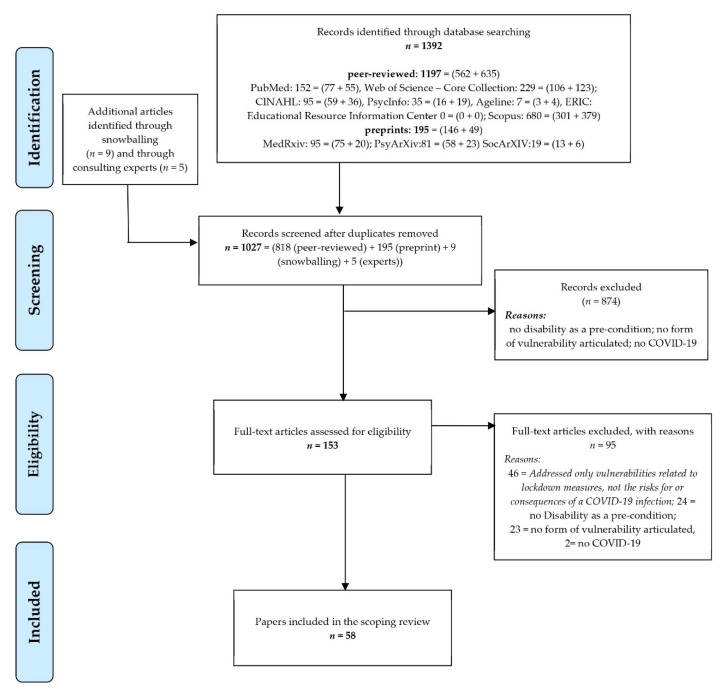
PRISMA flowchart of the scoping review with thematic analysis.

**Table 1 ijerph-18-04348-t001:** Quantitative map of the literature analyzed.

Characteristics	*n* (%)	Citations
**Publications Type And Source**		
Perspective papers (e.g., viewpoints, commentaries, essays, ethics/advocacy)	34 (58%)	[12,13,16,27,28,29,30,31,32,33,34,35,36,37,38,39,40,41,42,43,44,45,46,47,48,49,50,51,52,53,54,55,56,57]
Narrative summary/review (non-systematic)	3 (5%)	[58,59,60]
Framework or Experts recommendations	3 (5%)	[3,61,62]
Editorial or Letter to the editor	3 (5%)	[15,63,64]
**Non-empirical (peer-reviewed): SUB-TOTAL**	**41 (73 %)**	**-**
Cross-sectional surveys	4 (7%)	[65,66,67,68]
Systematic analysis of electronic medical/ administrative records	2 (4%)	[69,70]
Ecological study	2 (4%)	[71,72]
Systematic review	1 (2%)	[73]
Analysis of COVID-19 press conferences	1 (2%)	[74]
Analysis of webpages on accessibility compliance	1 (2%)	[75]
Documentary research and framework analysis	1 (2%)	[6]
Case report (4 patients)	1 (2%)	[76]
Empirical studies (peer-reviewed): SUB-TOTAL	13 (23%)	-
Ecological study	1 (2%)	[77]
Observational multicenter study	1 (2%)	[78]
Preprint studies: SUB-TOTAL	2 (4%)	-
**Geographical Focus**		
No geographical focus (e.g., applicable across locations)	37 (63%)	[3,12,13,15,28,29,31,32,33,34,35,36,37,39,41,42,43,44,46,47,48,49,50,51,52,53,54,55,58,60,62,68,69,71,75,76]
United States of America (USA)	9 (15%)	[27,56,57,59,67,70,72,73,77]
United Kingdom (UK)	5 (8%)	[16,40,45,61,78]
Latin America	2 (4%)	[6,64]
Low- and Middle-Income countries (LMICs)	1 (2%)	[74]
South Africa	1 (2%)	[38]
India	1 (2%)	[66]
Romania	1 (2%)	[30]
Italy	1 (2%)	[65]
**Health Conditions**		
People with disabilities, overall	32 (56%)	[3,6,12,13,15,16,31,32,34,35,38,39,40,43,44,47,48,49,50,51,53,54,56,57,59,63,67,72,73,74,75,77]
Adults with cognitive impairment (e.g., dementia) or intellectual disabilities	8 (14%)	[36,42,45,52,55,61,64,71]
People with disabilities living in residential or long-term facilities	3 (5%)	[30,70,78]
Spinal cord injury	3 (5%)	[37,58,68]
Children with disabilities (overall)	2 (4%)	[27,41]
Older adults experiencing disabilities	1 (2%)	[46]
People with developmental disabilities (overall)	1 (2%)	[69]
Visual impairments	1 (2%)	[33]
Autism spectrum disorder	1 (2%)	[28]
Cerebral palsy	1 (2%)	[29]
Cerebellar ataxia	1 (2%)	[62]
Down’s syndrome	1 (2%)	[76]
Severe mental illness	1 (2%)	[66]
Multiple sclerosis	1 (2%)	[65]
Rheumatoid arthritis	1 (2%)	[60]

## Data Availability

Not applicable.

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
