# Peer review of "Health Risks and Consequences of a COVID-19 Infection for People with Disabilities: Scoping Review and Descriptive Thematic Analysis"

_ijerph, 2021, doi:10.3390/ijerph18084348_

Round 1

Reviewer 1 Report

Authors of the manuscript entitled "Health risks and consequences of a COVID-19 infection for people with disabilities: Scoping review and descriptive thematic analysis" present an interesting and up-to-date topic, however, minor revisions must be made before manuscript may be considered for publication.

General

 Authors should carefully get familiar with Guidelines for Authors. Unfortunately the whole manuscript is written in inappropriate font and size (specifically, they should use Palatino Linotype font). Moreover, spacing seems to be too big.

Abstract

There is no need to specify components of the Abstract, such as background, methods etc.

Introduction

Introduction poses in-depth depiction of the issue, however, the aim of the study is not sufficiently emphasized at the end of this section. Please correct it.

Line 65-66 – It is obvious that authors abbreviated the phrase “People with disabilities” for brevity and convenience, so this sentence is unnecessary.

Methods

Was this review registered in the International Prospective Register of Systematic Reviews (PROSPERO) database or did the authors waive it?

Authors have chosen only published peer-reviewed studies, however, it is well-known that studies with negative results are less likely to be published. This limitation of potential bias should be mentioned in the Discussion section.

Results

All tables must be corrected, as in the current form they do not adhere to the Journal’s guidelines. Please look carefully at these guidelines.

Figure 1 – please provide the figure with better definition as currently is hardly readable.

Acknowledgements

Please adjust this section according to the guidelines for Authors (size and font)

References

Here Authors also should look on the Guidelines for Authors once again because references aren't presented in a proper way. For example, the title of the cited journal should be put as an abbreviated form, not full one.

In the first cited article “Redefining vulnerability in the era of COVID-19” the Authors are missing.

Author Response

We submit our response to this reviewer (labelled as reviewer # 1) in the document attached. The document also contains the response to the other reviewers for context.

Reviewer 2 Report

The paper certainly presents an interesting scoping review on the health risks or consequences of a COVID-19 infection for PwD. More clarity is needed in a number of areas. I have very minor suggestions which could help to clarify some aspects.

Introduction:

-Line 62: Historically, ‘ableism’ has dominated the way PwD have been perceived, including practices and discourses that exclude and discriminate against PwD: Meaning is not clear; please clarify.

-Line 69: “mostly due to the ableist structural, environmental factors that result in vulnerability”: please clarify.

-Lines 75-78: The hypotheses of the study need to be explained in detail. In the Discussion section, please analyze whether the results of the review support these initial hypotheses.

-The objective was “to synthesize the literature on the health risks or consequences of a COVID-19 infection for PwD”. In the Introduction section, please provide clear definitions of “health risks” and “consequences”.

Methods:

-Line 86: meaning of “descriptive thematic analysis" is not clear; please clarify.

-Line 101: “possible individual and social-level vulnerabilities to the effects of the COVID-19 pandemic”: please include examples.

Results:

-Figure 1: Please explain the meaning of the following information described in the flowchart: 818 + 195 + 9 + 5.

-Line 172: “Table 2 reflects the distribution of the analyzed articles”: this information is included in Table 1, not Table 2.

-Table 1: Down Syndrome should be included in the group of "intellectual disabilities".

-It is not appropriate to present p-values as follows: e.g. 3.3E-02.

-The Results section is very broad. It is recommended to reduce its length, highlighting only the main information of each of the studies analyzed.

-When the p-value is not significant, it is not necessary to indicate its exact value.

Discussion:

-Line 490 (conclusion): “This scoping review, addressing the initial stages of the COVID-19 pandemic, showed that PwD can experience disproportionate health risks and consequences from a COVID-19 infection”: In the Discussion section, it would be useful to highlight and discuss which are the most relevant health risks and consequences for this population group (priorities), as assessed by the authors based on the results of the review.

-Limitations: Please analyze what are the disadvantages that the non-inclusion of grey literature produces for the reader.

“No methodological quality assessments were performed”: Please analyze what are the disadvantages that the absence of this quality assessment produces for the reader (common in scoping review designs).

-Lastly, the paper would be significantly improved with the addition of specific recommendations for future research in the field studied.  Opportunities and directions of future work need to be highlighted more clearly.

Author Response

We attach a response to this reviewer's comments (here labelled as Reviewer # 2), but we do it so in the document that also has the responses to reviewer 1 and 3 for background. Thank you.

Reviewer 3 Report

Dear authors, good job, there are recommendations to improve the manuscript, please see attached document:

Pd. I forgot to put in the attached document, is it possible to reduce the words in the abstract, it is very long.

Author Response

We attach here our response to this reviewer's comments (here labelled as reviewer # 3), but we do it so in the document that contains the responses also to reviewer #1 and #2 for better context.

Thank you.
